# Circadian Rhythms in NLRP3 Inflammasome Regulation: Possible Implications for the Nighttime Risk of Gout Flares

Raewyn C. Poulsen [1,*] and Nicola Dalbeth [2]

[1]  Department of Pharmacology and Clinical Pharmacology, The University of Auckland, Auckland 1023, New Zealand
[2]  Department of Medicine, The University of Auckland, Auckland 1023, New Zealand; n.dalbeth@auckland.ac.nz
*  Correspondence: r.poulsen@auckland.ac.nz

**Abstract:** Gout flares more frequently start late at night or in the early morning compared to during the day. The reasons for this are unknown. Activation of the NLRP3 inflammasome in monocytes/macrophages is central to initiation of gout flares. Here, we review the mechanisms by which circadian clocks control the NLRP3 inflammasome and the implications of this for the nighttime pattern of gout flares. Several hormones involved in inflammation regulation, e.g., glucocorticoids, melatonin and melanocortins, are under circadian control, with both circulating hormone levels as well as the expression of their receptors on target tissues showing time-of day differences. In addition, the NLRP3 inflammasome is also under the control of the macrophage circadian clock, leading to time-of-day differences in expression of NLRP3 inflammasome components and susceptibility to inflammasome-activating stimuli. MSU crystal exposure leads to altered expression of circadian clock components in macrophages, leading to time-of-day-specific loss of repression of NLRP3 inflammasome activity. Taken together, there is clear evidence that circadian clocks regulate the NLRP3 inflammasome and that this regulation may be compromised by MSU crystal exposure in gout. Circadian control of the inflammasome may be one of the factors contributing to nighttime susceptibility to gout flares.

**Keywords:** circadian clock; NLRP3 inflammasome; gout flare; monosodium urate crystals

## 1. Introduction

Gout flares are painful, frequently unpredictable episodes of acute joint inflammation and are a major contributor to reduced quality of life in individuals with gout. Deposition of monosodium urate (MSU) crystals within a joint is the critical underlying risk factor for gout flares. However, why flares only occur sporadically despite the continual presence of MSU crystals within a joint remains incompletely understood.

It has long been recognized that gout flares more frequently initiate late at night or during the early hours of the morning rather than during the day. This is reflected in the classic description of gout written by Thomas Sydenham in 1683: "The victim goes to bed and sleeps in good health. About 2 o'clock in the morning, he is awakened by a severe pain in the great toe; more rarely in the heel, ankle or instep" [1]. In contemporary times, the number of Google searches for "big toe pain", a common symptom of gout flare, has been found to be highest in the early hours of the morning (3 a.m.–4 a.m.) [2], and epidemiological studies have shown that the risk of gout flare is 2.36 times higher during the early morning (12:00 a.m.–7:59 a.m.) and 1.26 times higher during the evening (4:00 p.m.–11:59 p.m.) than during the daytime (8:00 a.m. to 3:59 p.m.) [3].

The nighttime initiation of gout flare is intriguing given that it is now well established that immune function and inflammation are under circadian control. The purpose of this review is to discuss how circadian rhythms influence the activity of pathways known

to be involved in gout flare and how this may contribute to the nighttime risk of gout flare initiation.

## 2. Circadian Rhythms

By definition, a "circadian rhythm" is a rhythm that repeats approximately every 24 h, is sustained under constant conditions (i.e., can continue in the absence of any external stimulus) and is largely independent of temperature [4]. The first known description of circadian rhythms in biology was made by Jean-Jacques d'Ortous de Mairan in 1729, who observed that leaves of the Mimosa plant furled and unfurled over the course of 24 h and that they continued to do this even if the plant was maintained in constant darkness (constant conditions) [5]. Since this time, studies on laboratory animals, as well as a few ambitious studies on humans, have demonstrated that 24 h rhythms in rest/activity cycles continue even in constant darkness, indicating that they are generated by an endogenous mechanism rather than simply a response to external cues [6,7].

Although the presence of circadian rhythms have long been recognized, it was not until the end of the 20th century that the complex molecular circuitry or "circadian clock" driving circadian rhythms in cell function and organism activity began to be understood [8]. Since this time, circadian clocks have been identified in almost every cell in the body. Nucleated cells contain a highly conserved circadian clock comprising a series of self-regulating transcription and translation feedback loops (TTFLs) [9]. Known as the TTFL clock, this is the most well-studied type of circadian clock and the most relevant for immune cell function [10]. Anucleate cells which are devoid of the transcriptional machinery lack a TTFL clock. However, at least some anucleate cells contain an alternative type of circadian clock. For instance, anucleate mature red blood cells contain a peroxiredoxin/redox clock, and cyclical oxidation and reduction of peroxiredoxin drives circadian rhythms in red blood cell activity [11].

## 3. The Transcription/Translation Feedback Loop (TTFL) Circadian Clock

The TTFL clock comprises a series of regulatory feedback circuits. The core TTFL clock consist of the proteins ARNTL (more commonly referred to as BMAL1), CLOCK, Period and Cryptochrome. Humans express three different period proteins (PER1, -2 and -3) and two different cryptochromes (CRY1 and CRY2). BMAL1 is a transcription factor, and CLOCK has histone acetyl transferase activity. The two act as a heterodimer to promote the transcription of some genes, such as *PER* and *CRY*, but repress expression of others, e.g., *BMAL1* [12]. As a consequence of BMAL1:CLOCK-mediated transcriptional control, protein levels of PER and CRY subsequently increase, whereas levels of BMAL1 start to fall. PER and CRY also form a dimer, and together they repress the activity of BMAL1:CLOCK. This triggers a fall in *PER* and *CRY* transcription but removes repression of *BMAL1* transcription, thereby allowing BMAL1 levels to rise again. Other auxiliary feedback loops also act to regulate expression of the core clock components. For instance, reciprocal regulation between BMAL1 and the nuclear receptors REV-ERB ($\alpha$ and $\beta$) and ROR ($\alpha$, $\beta$ and $\gamma$) also controls *BMAL1* expression. Once activated, REV-ERB and ROR act as transcriptional regulators. Both recognize the same response element in target gene promoters but usually result in opposing effects on target gene transcription. For instance, REV-ERB represses *BMAL1* expression, whereas ROR promotes it [13]. A number of other feedback control mechanisms are also involved in the TTFL clock, with the end result that the abundance of clock components oscillates in a self-sustaining cycle [9]. One full oscillation of the cycle takes approximately 24 h. This cycle will continue in the absence of any external input (i.e., under constant conditions) but can also be influenced by rhythmic changes in the environment. This is a particularly important feature of circadian clocks, as it allows the cellular molecular clock machinery to synchronize with external circadian rhythms, such as the day/night cycle [14–17]. In doing so, the circadian clock provides a tool whereby cells and organisms can anticipate rhythmic changes in their environment and coordinate their activity appropriately. In complex multi-cellular organisms like humans,

there is a hierarchical structure to the circadian clock network, with a central clock located in the suprachiasmatic nucleus (SCN) of the hypothalamus contributing to the regulation of clocks in other "peripheral" tissues [18] (Figure 1).

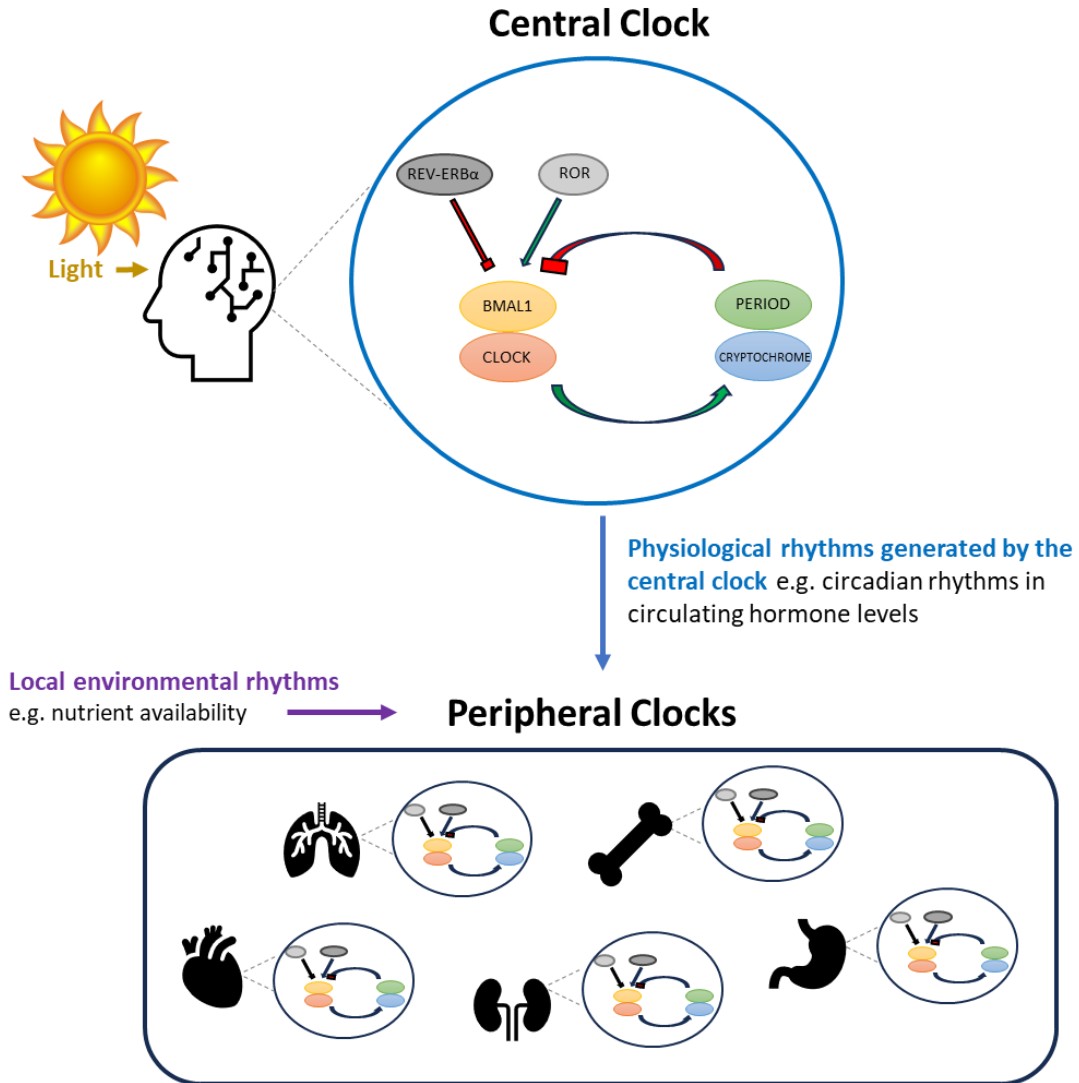

**Figure 1.** The hierarchical circadian clock network. In humans, the circadian clock network consists of a central clock located in the suprachiasmatic nucleus in the hypothalamus and peripheral clocks located in all other cells in the body. The central clock comprises a series of transcription and translation feedback loops (TTFLs). The core components of the circadian clock are the proteins BMAL1 and CLOCK and members of the PERIOD and CRYPTOCHROME families. BMAL1 and CLOCK dimerize and promote PERIOD and CRYPTOCHROME transcription. PERIOD and CRYPTOCHROME dimerize and inhibit the actions of BMAL1:CLOCK, resulting in the expression of BMAL1 and CLOCK oscillating in anti-phase with PERIOD and CRYPTOCHROME. Other auxiliary feedback loops also contribute to regulating this cycle, one of which is made up of the transcriptional regulator REV-ERBα and members of the ROR family. REV-ERBα represses BMAL1 expression, whereas ROR promotes it. The same TTFL machinery is present in nucleated cells throughout the body. Clocks in these tissues are termed "peripheral clocks". The central clock is directly regulated by light sensed through the eye, and this synchronizes its cycling with the day/night cycle. Most peripheral clocks are not light sensitive and instead are regulated by physiological rhythms generated by the central clock (such as circulating levels of hormones, e.g., glucocorticoids) as well as rhythms in the local tissue environment, e.g., nutrient availability driven by daily feeding/fasting cycles.

## 4. Central versus Peripheral Circadian Clocks

The central clock is directly regulated by light, and this enables it to sense the day/night cycle [15,19]. The central clock is comprised of a group of SCN neurons, each containing their own TTFL circuitry. Light, particularly in the blue wavelength at dawn, is sensed by specialized retinal ganglion cells located in the eye which contain the photopigment melanopsin [20,21]. This leads to activation of the retinohypothalamic tract which triggers a signaling cascade promoting *PER* transcription in neurons of the central clock [19]. In this way, cycling of the central clock TTFL becomes synchronized with the light/dark cycle. The TTFL clock components each control a number of processes within cells, and this therefore synchronizes SCN neuron activity with the light/dark cycle. One of the functions of the SCN is to control the release of hormones such as melatonin and corticosteroids, and, consequently, circulating levels of these hormones oscillate with a circadian rhythm synchronized to the day/night cycle.

Hormones, particularly corticosteroids, regulate peripheral clocks, and hence TTFL cycling in peripheral cells is partly controlled by the central clock [22]. However, peripheral clocks are also regulated by rhythms in other local environmental factors, and these can be more important for controlling peripheral clock cycling than cues from the SCN [23,24]. This was clearly demonstrated in mice, where peripheral clocks in tissues such as the liver and intestine were found to preferentially synchronize with rhythms in daily food availability rather than the light/dark cycle [24]. The extent to which the peripheral clock regulates cell activity varies between cell types [25]. However, most fundamental cell processes, such as the cell cycle, energy metabolism and cell differentiation, are consistently controlled by peripheral clocks across all cell types [26,27]. One of the mechanisms by which circadian clocks control cell activity in peripheral tissues is through the regulation of gene transcription. Several clock components (e.g., BMAL1:CLOCK) are direct transcriptional regulators that bind to specific DNA motifs in target genes to regulate their transcription [28]. However, the overall influence of the circadian clock on cell activity is more complex than just the direct control of gene expression. Clock components can compete with other transcriptional regulators to bind to similar DNA motifs; dimerize with other transcription factors, altering the profile of target genes expressed; and influence the epigenetic control of gene expression, e.g., by altering chromatin structure and/or miRNA expression [29,30]. The circadian clock is therefore one component of a collaborative network ultimately controlling the transcriptional and epigenetic landscape within a cell [30]. In addition, it is now widely recognized that the circadian clock has much broader effects in a cell than just the control of gene transcription. Both BMAL1 and PER2 regulate the activity of mTORC1, leading to circadian rhythms in protein translation independent of transcriptional activity [31,32]. In macrophages, circadian rhythms in RNA expression were found to be absent for 85% of all proteins showing circadian rhythms in abundance [33], highlighting that post-transcriptional mechanisms also have a crucial role in clock-mediated control of cell activity.

## 5. Circadian Rhythms in Immune Function and the NLRP3 Inflammasome

Circadian rhythms in immune function were first demonstrated experimentally in 1960, when it was shown that a dose of Escherichia coli endotoxin that was highly lethal to mice when administered at one time of day was well-tolerated if administered 8–12 h earlier [34]. This difference in endotoxin susceptibility followed a predictable 24 h pattern, strongly indicating the presence of circadian rhythms in immune function. Since this time, phagocytosis and the anti-microbial activity of neutrophils [35], antibody production and response to vaccination [36,37], and immune cell numbers (including T-cells and leukocytes [38,39]) have all been shown to oscillate with a circadian rhythm. Of specific relevance to gout, the activity of the NLRP3 (Nod-like receptor protein 3) inflammasome, a pivotal pathway driving gout flare [40], has also been shown to be regulated by the central clock through the actions of clock-controlled hormone release as well as by peripheral clocks within immune cells [41,42].

## 5.1. The NLRP3 Inflammasome

The NLRP3 inflammasome comprises three proteins: NLRP3, ASC (apoptosis-associated speck-like protein containing a caspase recruitment domain) and Caspase 1. Activation of the NLRP3 inflammasome occurs by a two-step process involving first priming of the inflammasome, leading to production of the inflammasome components, and then assembly of the components into a functional form.

### 5.1.1. NLRP3 Inflammasome Priming

Toll like receptors (TLRs) are one of the major types of sensors that initiate NLRP3 inflammasome priming. TLRs are abundantly expressed by a wide range of immune cells [43] as well as many non-immune cells, e.g., chondrocytes. The main role of TLRs is to sense pathogens or tissue damage, and they do this by utilizing molecules produced as a result of either pathogen attack or tissue damage (collectively known as pathogen-associated or damage-associated molecular patterns (DAMPs and PAMPs)) as their agonists. Humans express ten functional TLRs which are activated by different DAMPs and PAMPs, leading to activation of NF-κB (nuclear factor κB), a transcription factor which primes the inflammasome by promoting transcription of *NLRP3* and *pro-IL-1β*. In addition, inflammasome priming can also occur through TLRs and/or NF-κB-independent mechanisms, and a number of transcriptional regulators also directly or indirectly control *NLRP3* and *pro-IL-1β* expression [44] (Figure 2).

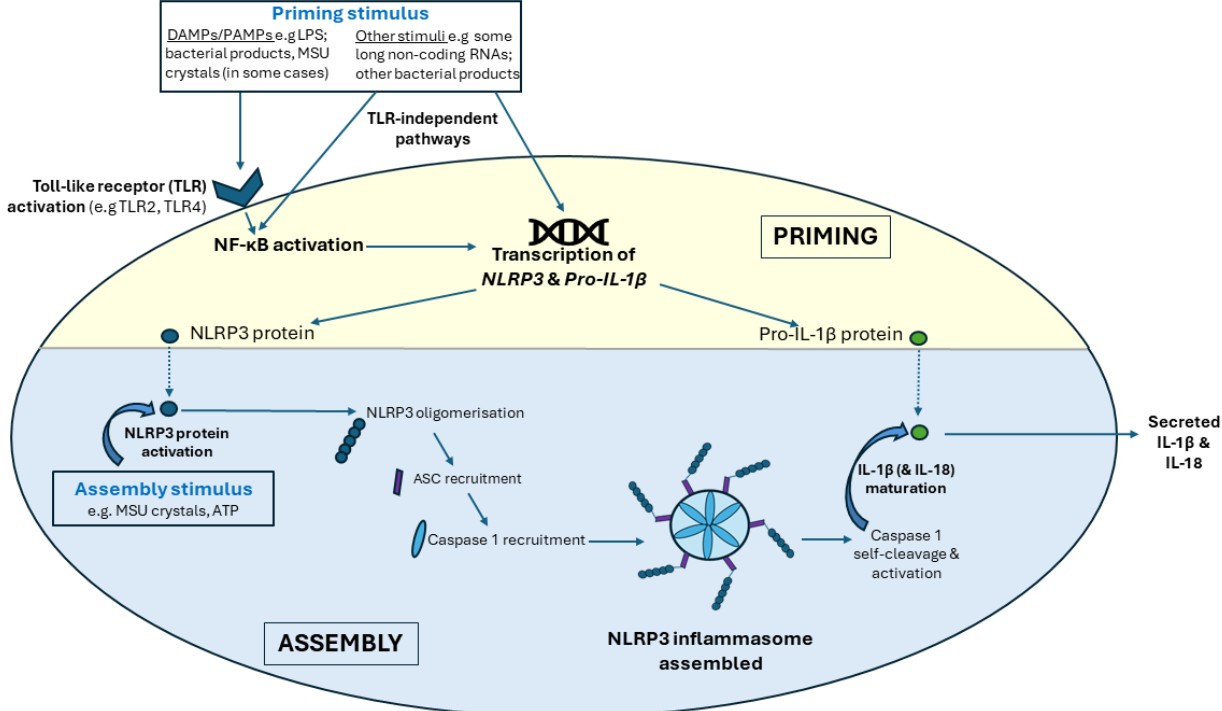

**Figure 2.** NLRP3 inflammasome activation. NLRP3 inflammasome activation occurs by a two-step process involving, first, priming, then assembly of the inflammasome. Priming stimuli include pathogen-associated and damage-associated molecular patterns (DAMPs/PAMPs) which act on toll-like receptors. Although MSU crystals can serve as DAMPs and reliably activate inflammasome priming in vitro, this does not always occur in vivo and instead other DAMPs/PAMPs stimulate the priming step. Toll-like receptor activation leads to activation of nuclear factor kappa B (NF-κB) and upregulation of *NLRP3* and *Pro-IL-1β* gene transcription. NLRP3 inflammasome priming can also be stimulated by pathways independent of TLR activation. A range of other regulators also contribute to controlling *NLRP3* and *Pro-IL-1β* expression. The assembly step in the NLRP3 inflammasome

activation process requires a second stimulus. MSU crystals as well as a number of other factors, such as ATP, can stimulate the assembly step. This leads to activation and oligomerization of NLRP3 protein, followed by recruitment of ASC and caspase 1, resulting in NLRP3 inflammasome assembly. Assembly of the inflammasome triggers caspase 1 activation by self-cleavage. Active caspase 1 drives the maturation and secretion of IL-1β and IL-18.

### 5.1.2. NLRP3 Inflammasome Assembly

The second step in NLRP3 inflammasome activation is triggered by activation of the NLRP3 protein. NLRP3 serves as an intracellular sensor and can be activated by a large range of stimuli, such as ATP, bacteria/viruses and microcrystalline deposits, including MSU [45–47]. Most NLRP3-activating stimuli have been shown to induce reactive oxygen species (ROS) generation, particularly mitochondrial ROS, and this is a critical step in NLRP3 activation [45–48]. Once an NLRP3-activating stimulus is received, NLRP3 proteins oligomerize and recruit ASC proteins which form filaments radiating inwards from the NLRP3 oligomers like spokes on a wheel [49]. Caspase 1 is then recruited to the NLRP3/ASC structure, and this triggers caspase-1 activation by self-cleavage. Caspase 1 is the effector of the inflammasome, driving maturation and subsequent secretion of the inflammatory cytokines IL-1β and IL-18 (Figure 2). Cleaved (active) caspase 1 is much less stable than the full-length inactive protein and is therefore more rapidly degraded. This means that caspase-1 activation, and therefore NLRP3 inflammasome activity, is self-limiting [50].

### 5.1.3. NLRP3 Inflammasome Activation in Gout Flares

In vitro, MSU crystals can stimulate both the priming and assembly steps in inflammasome activation. MSU crystals activate both TLR2 and TLR4 on monocytes and macrophages, and MSU crystals are a reliable stimulator of NLRP3 inflammasome activation in these cells in culture [40,51]. However, in vivo, exposure to MSU crystals alone is often insufficient to trigger both priming and assembly of the inflammasome, as evidenced by the sporadic nature of gout flares. Rather, the presence of MSU crystals in a joint creates an environment that facilitates inflammasome assembly, but a second trigger is often required to initiate inflammasome priming [52,53]. Various triggers for the priming step have been proposed, including certain foods/nutrients, such as free fatty acids and alcohol (reviewed in [54]). It is noteworthy that many of the proposed food triggers for inflammasome priming, e.g., alcohol, are often more frequently consumed in the evening rather than the morning, and this may be one of the factors contributing to the nighttime risk of gout flare initiation. Changes in serum urate levels, e.g., as a consequence of urate-lowering therapy, are also associated with increased risk of gout flare [55,56]. Several studies have demonstrated that the fractional excretion of urinary urate is lower at night [57,58], whereas serum urate levels have been found to be highest in the early hours of the morning (3 a.m.–9 a.m.) [59], falling through the day [59–61]. Although there is some evidence of possible circadian involvement in control of urate levels (e.g., studies in mice have shown that xanthine oxidase (which catalyzes the conversion of hypoxanthine to xanthine and xanthine to uric acid) is indirectly under circadian control in the liver as a consequence of regulation by bile acid [62]), current data suggest that diurnal factors, such as day/night patterning of food intake, rather than circadian rhythms may be the major driver of day/night differences in urate levels. This was elegantly demonstrated by one study which showed that day/night patterning in urinary urate excretion was lost when individuals were accommodated in an isolation unit with food intake spread evenly over the full 24 h period ("constant conditions") [58]. There is considerable interindividual variability in the extent of day/night differences in urate levels. For instance, one study found that the change in day versus night serum urate levels ranged from 0 to 39 μM in otherwise healthy young adults, with a similar change (10–37 μM) found in young adults with diabetes; however, in older adults (60–80 years) with hypertension, no significant change in day/night serum urate levels was detected [60]. Both the type and extent of

diurnal factors that may influence the risk of NLRP3 inflammasome activation and gout flares are likely to differ between individuals, since diurnal rhythms are driven by daily routines which are governed by lifestyle and behavior. Aside from diurnal factors, a second consideration for understanding time-of-day susceptibility to NLRP3 inflammasome activation is circadian control of immune function, which results in the extent to which the immune system responds to stimuli varying over the day.

## 6. Mechanisms for Circadian Control of NLRP3 Inflammasome Activity: Role of the Central Clock

One of the mechanisms by which the central clock contributes to controlling immune function is through the regulation of hormone levels [63,64]. Due to central clock control, circulating melatonin levels show a pronounced peak at night, whereas production of the glucocorticoid cortisol peaks in the morning around the time of wakening. Circadian rhythms in melatonin and cortisol have been linked with circadian rhythms in inflammatory cytokine levels in blood as well as circadian rhythms in disease symptoms, such as early morning stiffness and pain in rheumatoid arthritis [63,64]. It is therefore possible that circadian rhythms in melatonin and cortisol also contribute to time-of-day susceptibility to gout flares.

Many other endocrine systems are also regulated in a circadian manner, one of which is the melanocortin system. The melanocortins are a group of seven peptide hormones derived by post-translational cleavage of the pre-prohormone proopiomelanocortin (POMC) [65]. The melanocortins include adrenocorticotropic hormone (ACTH) and melanocyte-stimulating hormones (MSHs), which have been implicated in controlling inflammation in gout. Here, we will focus on circadian rhythms in melatonin, glucocorticoids and the melanocortins, ACTH and MSHs, and on the potential impact of these for gout flares.

### 6.1. Melatonin

The major source of circulating melatonin is the pineal gland, and pineal melatonin secretion is directly regulated by the central clock. Several peripheral tissues, including the skin, the retina, the thymus and the gastro-intestinal tract, as well as lymphocytes and macrophages, also produce melatonin, which can act through autocrine or paracrine mechanisms. Production of melatonin by the retina follows a circadian rhythm regulated by the light/dark cycle [66]. However, melatonin production is not circadian in other non-pineal tissues and instead is initiated in response to specific stimuli [67]. In macrophages, NF-κB promotes melatonin synthesis [68].

Melatonin acts through both receptor-dependent and -independent mechanisms. It can directly scavenge reactive oxygen species (ROS) and reactive nitrogen species (RNS) and therefore has antioxidant activity independent of receptor binding [69]. It can also directly or indirectly activate several different receptors, including melatonin receptors 1 and 2 (MT1 and MT2) and the circadian clock associated nuclear receptor RORα [70–72]. mRNA expression of *MT1* has been shown to follow a circadian rhythm [73], and the activity of ROR-α is also under circadian control. This suggests that the profile of receptors available for melatonin-mediated activation varies over the day.

MT1, MT2 and RORα are widely expressed by immune cells, and melatonin has been shown to both promote and inhibit inflammation depending on melatonin dose as well as the cellular context [74] (reviewed in [75]). Given the circadian rhythm in the abundance/activity of melatonin receptors and that each receptor activates different signaling cascades [76], it is likely that the time of day of exposure also influences the outcome of melatonin signaling. This may be particularly relevant for understanding the outcome of localized non-circadian increases in melatonin production in the inflammatory response in cells such as macrophages.

Melatonin, the NLRP3 Inflammasome and Gout

When used at high pharmacological concentrations, melatonin has been shown to repress NLRP3 inflammasome activity. For instance, treatment with melatonin suppressed NLRP3 inflammasome activity following radiotherapy in a murine cancer model [77,78] and led to reduced levels of NLRP3 as well as active IL-1β in bone marrow stem cells of mice post-ovariectomy [79]. In relation to gout, melatonin was found to abrogate the increase in paw thickness; inflammatory cell infiltration; and *il-1β*, *nlrp3* and *il-6* expression following injection of MSU crystals into mouse paws and to suppress IL-1β and IL-6 production following MSU crystal/lipopolysaccharide (LPS) treatment of mouse peritoneal macrophages in vitro [80].

However, these studies were designed to test the efficacy of melatonin as a treatment rather than to test the effects of endogenous melatonin, and high doses of melatonin were used over prolonged periods (e.g., localized application of 45 mg melatonin/day to oral mucosa in vivo [77]). For context, the maximum plasma level of endogenous melatonin in humans is approximately 60–70 pg/mL, reducing to <10 pg/mL during the day [81]. Whether physiological concentrations of endogenous melatonin can inhibit NLRP3 inflammasome activity is unclear.

Given findings of an inhibitory effect of exogenous melatonin on NLRP3 inflammasome activity, it is interesting that one study has reported that plasma melatonin levels are lower in patients during a gout flare compared to people with intercritical gout or healthy individuals [80]. Although this is intriguing, the interpretation of this finding is limited, as melatonin was only measured at a single timepoint which corresponded to a time when melatonin levels are naturally very low (9–11 a.m.). The mean melatonin concentration was 8.0 pg/mL in individuals experiencing a gout flare, 9.6 pg/mL in those with intercritical gout and 10.66 pg/mL in healthy controls [80], indicating that sampling occurred around the nadir in melatonin cycling [81]. Compared to the nighttime peak melatonin concentration of 60–70 pg/mL [81], the magnitude of the difference between groups in this study is extremely small and the difference at this specific timepoint is unlikely to be physiologically significant. Determining whether more pronounced differences occur during the hours in which melatonin levels are normally high or whether there is a shift in the timing of peak melatonin levels in individuals with gout (which could also cause a difference in melatonin concentrations when measured at a single timepoint) is necessary in order to understand the implications of this finding.

*6.2. Glucocorticoids*

Similar to melatonin, both endogenous production of glucocorticoids as well as the sensitivity of target tissues to glucocorticoid stimulation are regulated by the circadian clock. In addition to circadian control, glucocorticoid synthesis is also increased in times of stress. Glucocorticoids can signal through both receptor-dependent and receptor-independent mechanisms. The expression and the activity of the glucocorticoid receptor (GR) is under circadian control, and this can lead to circadian variation in the outcome of glucocorticoid exposure [82].

The GR is an intracellular receptor which, upon activation, is trafficked to the nucleus, where it regulates the transcription of target genes. This includes feedback inhibition of its own expression. As a consequence, glucocorticoid receptor levels in target tissues are relatively high at the time of day when circulating endogenous glucocorticoid levels peak, but this is followed by a subsequent fall in receptor levels [82]. The GR also physically interacts with circadian clock components, including REV-ERBα, CRY1, CRY2 and BMAL1/CLOCK, and this alters the affinity of GR for DNA binding [83–85]. Interaction with BMAL1/CLOCK reduces GR-mediated gene transcription [84], whereas CRY1, CRY2 and REV-ERBα alter the recruitment of GR to target genes, changing the profile of GR-mediated gene expression [83,85]. This has largely been studied in the context of energy metabolism, and interaction with clock components has been shown to result in striking differences in GR-mediated control of energy metabolism in the liver [83] and in the profile

of genes up- or downregulated by the GR in hepatic cells [85]. It is therefore likely that interaction with clock components also influences GR-mediated control of immune function and inflammation, leading to time-of-day differences in the outcome of GR signaling. In possible support of this, the ability of a single concentration of hydrocortisone to repress expression of TNF-α in peripheral blood mononuclear cells (PBMCs) has been shown to vary depending on the time of day, with the strongest repression in the morning [86].

Glucocorticoids, the NLRP3 Inflammasome and Gout

Glucocorticoids are normally associated with anti-inflammatory effects, and therefore the nighttime dip in circulating glucocorticoid levels has been proposed as a contributing factor to the nighttime risk of gout flare [3]. The ability of glucocorticoids to dampen inflammation during gout flare is demonstrated by the efficacy of pharmacological synthetic glucocorticoids for managing gout flares [87]. However, the actions of glucocorticoids are complex. Glucocorticoids appear to act downstream of NLRP3 inflammasome activation, repressing pro-inflammatory cytokine production, e.g., through inhibiting activity of AP-1 and NF-κB [88,89]. However, glucocorticoids upregulate expression of *TLR2*, *TLR4* [90,91] and *NLRP3* [42]. This suggests that while they dampen existing NLRP3 inflammasome activity, they may also promote NLRP3 inflammasome priming. This is consistent with other findings that show that while glucocorticoids repress expression of some inflammatory mediators, they also promote expression of others, e.g., chemokines and complement proteins, suggesting that they act to inhibit existing inflammation but sensitize the immune system so it is better prepared to react to a subsequent stimulus [90].

It is currently unknown how long after inflammasome priming symptoms of gout flare manifest. This is an important consideration for understanding whether the stimulatory effects of endogenous glucocorticoids on *TLR* and *NLRP3* expression could have significance for gout flares. Aside from glucocorticoids, a multitude of other factors activate or repress NLRP3 priming; some of these factors are also under circadian control. For instance, NF-κB activity (promoter of NLRP3 priming) has been shown to increase in human endothelial cells from the latter part of the day/early evening [92], whereas protein levels of the Aryl hydrocarbon receptor (AhR) (a repressor of NLRP3) were found to be lowest in multiple peripheral tissues in rats in the latter part of the light period (day) [93]. It is possible that the effects of glucocorticoids, acting in concert with other regulators of NLRP3 priming, facilitate inflammasome priming during the day, sensitizing the inflammasome to a stimulus. In contrast, the nighttime dip in glucocorticoid levels leads to loss of the repressive effects of glucocorticoids on inflammatory cytokine production from the activated inflammasome. Therefore, glucocorticoids may act by more than one mechanism to contribute to the nighttime risk of gout flare symptom onset.

Aside from circadian rhythms in endogenous glucocorticoid levels, findings in hepatic tissues of marked differences in GR expression and the profile of GR-controlled target genes over the course of a day raise the possibility that the responses of immune cells to stress-induced upregulation of glucocorticoid production may differ depending on the time of day of exposure. This is potentially relevant, as in some situations, e.g., acute stress, when the immune system is already challenged, the overall outcome of glucocorticoid exposure is pro-inflammatory, not anti-inflammatory [94]. Stress has been identified as a risk factor for gout flare [54]. It is possible that circadian rhythms in GR availability and signaling result in time-of-day differences in susceptibility to the NLRP3 inflammasome following stress exposure.

### 6.3. Melanocortins: ACTH and MSHs

ACTH as well as all three MSH peptides are produced in the hypothalamus by enzymatic cleavage of the POMC prohormone [95]. Circulating levels of ACTH, β-MSH and γ-MSH (but perhaps not α-MSH [96]) follow a circadian rhythm [97–99]. Levels of ACTH are lowest in the early evening in humans [98], and levels of both γ-MSH and β-MSH have been shown to follow a similar rhythm to ACTH, with an evening dip in

circulating levels [98,99]. However, there is some indication that circadian rhythms in MSHs may be influenced by season [98]. One study demonstrated that circulating levels of β-MSH followed a similar rhythm to ACTH in summer, with levels highest in the morning and lowest in the early evening. However, while the same circadian rhythm in ACTH was maintained in winter, the reverse patterning was evident for β-MSH, with circulating β-MSH levels peaking at the time of the early evening nadir in ACTH levels [98]. To our knowledge, this potential seasonal difference in the rhythmicity of MSHs has not been further explored. In the context of gout, such differences could be of relevance given findings of seasonal variation in the risk of gout flares [100].

ACTH and all three MSHs act through melanocortin receptors. There are five melanocortin receptors (MC1R-MC5R), which vary in their relative affinities for the melanocortin peptides and their tissue distribution. Whereas MC2R solely binds to ACTH, MSH peptides are also agonists for the other four receptors [95]. Melanocortin receptor signaling is integrally involved in regulating circadian rhythms in various physiological processes. For instance, activation of melanocortin receptors in the adrenal cortex contributes to the control of circadian rhythms in circulating cortisol levels [101]. ACTH is also upregulated in times of stress and contributes to stress-induced increases in cortisol levels. Melanocortin signaling also has a major role in regulating energy metabolism and satiety. Although this has mostly been studied in relation to melatonin receptor signaling in neural tissue, MC3R has been shown to contribute to synchronizing peripheral clock cycling in the liver with rhythms in feeding/fasting cycles, indicating that melatonin receptor signaling also has a vital role in controlling circadian rhythms in peripheral tissues [102]. To our knowledge, it is unknown whether expression of melanocortin receptors on target tissues also varies across a day, but this would be interesting to determine (particularly for MC3R) given that circadian rhythms in both the glucocorticoid receptor and melatonin receptors have been observed and that these are also hormones involved in transmitting central clock signals to peripheral tissues. In the context of gout, this could be particularly relevant given that MC3R expression on immune cells such as monocytes also been shown to be involved in MSU crystal-induced inflammatory signaling [103,104].

ACTH, MSHs, the NLRP3 Inflammasome and Gout

Several studies have demonstrated that treatment with ACTH is efficacious in the management of gout flares [105–108]. Although this was largely presumed to be due to the role of ACTH in stimulating production of glucocorticoids, studies in rats demonstrated that a dose of ACTH which had no effect on circulating corticosterone levels was still able to suppress MSU crystal-induced inflammation, indicating that ACTH can exert anti-inflammatory effects independently of corticosteroids [104]. The anti-inflammatory effects of ACTH were found to be dependent on MC3R [104]. Consistent with this, other MC3R agonists, including α-MSH and the MC3R selective agonist γ2-MSH, have also been shown to exert anti-inflammatory activity [103,104,109], and this appears to be at least partially due to direct actions on monocytes [109]. For instance, treatment of MSU crystal-exposed monocytes with α-MSH resulted in reduced production of the inflammatory cytokines IL-1β, TNF-α and IL-8 but had no effect on caspase 1 activity. It also resulted in reduced neutrophil migration but had no effect on TLR2 or TLR4 expression on neutrophils [109]. This suggests that α-MSH does not inhibit NLRP3 inflammasome activation but instead represses anti-inflammatory cytokine production downstream of NLRP3 inflammasome activation, potentially contributing to resolving NLRP3 inflammasome-induced inflammation. Again, these studies were designed to test the efficacy of pharmacological MSH/MC3R agonists and high doses were used, e.g., 1 μM for in vitro experiments [109] and 9.6 nmol administered subcutaneously or intra-articularly for in vivo studies [103]. In comparison, morning plasma level in humans have been measured at 1.0 nM for α-MSH [110], up to 240 pM for β-MSH [98], 6.4 nM for γ-MSH [99] and approximately 20–40 pM for ACTH [98]. Whether endogenous concentrations of MSH/ACTH are sufficient to also exert anti-inflammatory activity and therefore whether the morning peak in circulating

ACTH/MSH levels contributes to time-of-day differences in inflammatory responses is unknown.

### 6.4. Other Mechanisms by Which the Central Clock May Influence Gout Flares

The central clock also drives circadian rhythms in other aspects of physiology and behavior which could potentially contribute to risk of gout flares. For instance, body temperature is lowest at night, and lower temperatures have been shown to enhance MSU crystal-induced NLRP3 inflammasome activation in macrophages in vitro [111]. Circadian rhythms in monocyte/macrophage cell numbers may also contribute to time-of-day differences in NLRP3 inflammasome activation, and, in humans, leucocyte numbers are higher during the night than during the day [112]. Phagocytic activity of immune cells is also under circadian control, peaking during the hours of light exposure in mice, the "rest period" in these nocturnal animals [113]. Although it is likely, therefore, that phagocytosis is also under circadian control in human cells, to our knowledge, it is currently unknown whether phagocytosis peaks during the day or night in humans. This is potentially relevant to gout flares, however, as phagocytosis of MSU crystals has a key role in NLRP3 inflammasome activation. Overall rest–activity rhythms also likely contribute to the nighttime risk of gout flare. For instance, dehydration (for which there is an increased risk during sleep at night) [114] and sleep apnea [115] have been identified as risk factors for gout flares.

## 7. Mechanisms for Circadian Control of NLRP3 Inflammasome Activity: Role of Peripheral Clocks

In addition to the central clock, peripheral clocks within monocytes and macrophages also control both the expression and activity of various components of the NLRP3 inflammasome activation pathway. There is also emerging evidence that the gouty joint environment and/or MSU crystals specifically cause changes in the circadian clock within macrophages [41,116], potentially leading to loss of normal circadian repression of the NLRP3 inflammasome [41].

### 7.1. Peripheral Clock Control of NLRP3, TLRs and NF-κB

Circadian rhythms in both the expression of toll like receptors (TLRs) and the responsiveness of TLRs to their respective ligands have been identified in mice; however, there are differences in the TLRs that are under circadian control between different cell populations [117].

In adherent splenocytes isolated from mice at various intervals over a 24 h period, circadian rhythms in expression of all *tlr* except *tlr2* were evident. Interestingly, despite the lack of oscillation in *tlr2* expression, the response to Tlr2 agonists was found to differ depending on the time of day of exposure. RNA levels of *il1* and *il6* were higher following exposure to Tlr2-specific agonists in cells isolated from mice 1 h after lights on compared to cells isolated 1 h after lights off, indicating that murine splenocytes have a greater sensitivity to Tlr2 agonists during the hours of light exposure. In contrast, the response of Tlr4 to its agonist, LPS, was highest in cells isolated during the hours of darkness [117].

Adherent splenocytes are a mix of several cell types, including macrophages. When a macrophage subpopulation was specifically examined, RNA levels of *tlr2, -4* and *-6* were found to be lower in samples collected from animals during the hours of light exposure than in samples collected during the hours of darkness, whereas *tlr1, -3, -5, -7* and *-8* remained relatively constant across 24 h. This is consistent with previous findings that the profile of genes under circadian control differs by cell type/between tissues. Macrophages may therefore show a different pattern of agonist responsiveness compared to adherent splenocytes.

Similar to findings with *tlr2* and *tlr4*, expression of *nlrp3* has also been shown to be under peripheral circadian clock control [41,118,119], and in mice, the peak in circadian *nlrp3* expression was found to occur during the hours of darkness (the active period for nocturnal mice) [118,119]. This time-of-day patterning in *nlrp3* expression has been shown

to be due to direct regulation by the circadian clock-associated transcriptional repressor REV-ERBα [118,119]. The *nlrp3* promoter contains a Rev-erbα response element [118], and Rev-erbα has been shown to repress *nlrp3* expression in the colon of mice, the human THP-1 macrophage-like cell line and peritoneal primary murine bone marrow-derived macrophages, as well as primary human PBMC-derived macrophages [41,118,119]. BMAL1 has also been shown to repress *NLRP3* expression, although the mechanism involved is unclear [41]. In contrast, ROR-γ response elements have been detected in the promoter regions of both Nlrp3 and Il-1β, and ROR-γ has been shown to promote expression of both [120].

Rev-erbα was also found to directly repress transcription of *RelA* (encoding the p65 subunit of NF-κB) in murine tissue, contributing to *NLRP3* repression [118]. NF-κB has also previously been shown to be regulated by CLOCK. Direct protein–protein interactions between CLOCK and the p65 NF-κB subunit lead to increased NF-κB-mediated transcriptional activation [121]. CLOCK is unusual among circadian clock components in that 24 h rhythms in CLOCK expression do not occur in all cell types. However, binding of CLOCK to BMAL1 prevents binding of CLOCK to p65; therefore, regardless of whether the abundance of CLOCK protein changes over the course of a day, the availability of CLOCK for RelA binding is likely to still show circadian variation. Although differences between cell types can occur, BMAL1 protein levels are often higher during the day than the night.

Taken together, findings from these studies suggest that the propensity for NLRP3 inflammasome activation may be highest at night in mice due to higher expression of Tlrs, Nlrp3 and expression/activity of p65 NF-κB. While these findings are intriguing, whether the same also occurs in humans remains to be determined. Some circadian rhythms show the same day/night patterning in nocturnal mice compared to diurnal humans, whereas others show the reverse patterning [122]. For instance, circulating melatonin levels show the same patterning in mice and humans, with peak melatonin levels at night in both, yet circulating corticosteroid levels are highest at dusk in nocturnal rodents and highest at dawn in diurnal animals [123]. Immune cell numbers also show opposite patterning in nocturnal versus diurnal animals, with the number of circulating leukocytes peaking at night in humans but during the day in mice [112]. The magnitude of MSU crystal-induced upregulation of caspase-1 activity and IL-1β production in human THP-1 macrophages was found to differ depending on the time of day of MSU crystal exposure [41]. Although it is not possible to correlate findings from this in vitro study with "day" and "night", higher activation was predicted to correlate with night based on the cycling of circadian clock protein component expression in the cells.

### 7.2. Emerging Evidence for Disruption of the Macrophage Peripheral Circadian Clock in Gout

A key feature of circadian clocks is their ability to sense rhythms in their environment and to adjust their cycling accordingly. It is perhaps not surprising that changes in the circadian clock often occur in disease states. Changes in the relative abundance of specific clock components within cells of diseased tissue have been observed in a wide range of pathologies, including multiple forms of cancer, neurodegenerative diseases such as Alzheimer's and Huntington's, as well as arthropathies, including rheumatoid arthritis and osteoarthritis [124–127]. Emerging evidence suggests this may also be the case in gout.

By gene ontology analysis of MSU crystal-exposed murine bone marrow-derived macrophages, circadian clock genes were found to be differentially regulated compared to PBS-treated controls, indicating that MSU crystal exposure results in changes in expression of components of the macrophage circadian clock [116]. Although the consequences of these changes were not investigated, this study also demonstrated that MSU crystal exposure resulted in widespread changes in expression of genes involved in energy metabolism as a result of persistent upregulation of JUN and the AP-1 transcription factor, and this contributed to inflammation. This is interesting, as energy metabolism is heavily regulated by the circadian clock and almost all enzymes involved in all energy metabolism pathways show circadian rhythms in expression. In addition, both expression of JUN and the binding

activity of API-1 are under circadian control [128]. It is possible that MSU crystal-induced changes in the macrophage clock contributed to the mechanism by which MSU crystals upregulated inflammation in this model. In support of this, MSU crystal exposure has also been shown to induce changes in expression of circadian clock components in the human THP-1 macrophage cell line. In particular, protein levels of both BMAL1 and REV-ERBα, two known inhibitors of NLRP3 inflammasome activation and/or inflammatory cytokine production [118,129], were reduced following MSU crystal exposure [41]. Both BMAL1 and REV-ERBα were shown to regulate NLRP3 inflammasome activity, and knockdown or pharmacological blockade of either resulted in increased MSU crystal-induced caspase1 activity and IL-1β production, indicating increased NLRP3 inflammasome activation [41].

Interestingly, in THP-1 macrophages not stimulated with MSU crystals, REV-ERBα protein levels were found to be relatively constant across a 24 h period in THP-1 macrophages, but protein levels of BMAL1 showed marked differences, with high expression for approximately 12 h followed by negligible expression for the following 12 h [41]. The effect of MSU crystals on REV-ERBα expression showed time-of-day dependence with a greater level of repression at the timepoints when BMAL1 levels were also naturally low and less repression when BMAL1 levels were naturally high [41]. This meant that in MSU crystal-exposed THP-1 macrophages, both BMAL1 and REV-ERBα levels were very low for an approximately 12 h period coinciding with the period when BMAL1 levels are naturally low [41]. Although it is difficult to extrapolate circadian timing from in vitro studies to "day" and "night", BMAL1 protein levels have been shown to be lowest during the night in other studies, suggesting that MSU crystal exposure leads to a particularly pronounced loss of circadian clock-mediated repression of NLRP3 inflammasome activation at night. Similarly, the level of caspase 1 activity and the amount of IL-1β secretion from THP-1 macrophages was found to be higher when cells were exposed to MSU crystals at the time of day when BMAL1 was naturally lower and there was greatest MSU crystal-induced repression on REV-ERBα protein levels. Although this study focused on BMAL1 and REV-ERBα only, MSU crystal exposure has been shown to result in altered expression of other circadian clock components as well. Notably, CLOCK was found to be upregulated in both human monocyte-derived macrophages and murine bone marrow-derived macrophages following MSU crystal exposure [116]. Given that CLOCK can promote NF-κB activation in a time-of-day-dependent manner [121], this may also be relevant for understanding the time-of-day differences in susceptibility to NLRP3 inflammasome activation.

Taken together, findings from these studies suggest that the presence of MSU crystals in a joint could lead to changes in expression of circadian clock components within the resident macrophages, leading to loss of normal circadian clock repression of NLRP3 inflammasome activity. This may lead to altered sensitivity of macrophages to NLRP3 inflammasome-activating stimuli, with the potential implication that a stimulus which would have limited NLRP3 inflammasome-activating ability if encountered at one time of day could lead to marked NLRP3 inflammasome upregulation if encountered at another.

## 8. Summary of the Effect of Circadian Clocks on the NLRP3 Inflammasome

Current evidence suggests that central and peripheral clocks may contribute to controlling the NLRP3 inflammasome at two levels. Firstly, they may contribute to regulating inflammasome priming through controlling expression of NLRP3 pathway components. This has largely been demonstrated in murine macrophages (summarized in Table 1). Secondly, they may control the extent of inflammatory response generated due to inflammasome activation by regulating downstream inflammatory cytokine production, and this has largely been demonstrated in studies with human macrophages (summarized in Table 2). However, while the day/night patterning of some of the proposed clock-mediated effects on the NLRP3 inflammasome and inflammation potentially align with an increased nighttime risk of gout flares, this is not the case for all proposed clock-mediated effects (highlighted in italics in Tables 1 and 2). In particular, the opposite day/night patterning for the peaks in circulating glucocorticoid levels in mice compared to humans raises questions

about whether endogenous glucocorticoids promote NLRP3 inflammasome priming in gout. Differences between mice and humans in the patterning of circulating hormone levels also raise questions about the relative contribution of endogenous levels of glucocorticoids, melatonin and/or melanocortins in regulating inflammation generated by the NLRP3 inflammasome, especially given that endogenous concentrations of these hormones are substantially lower than the concentrations at which these hormones have been shown to have anti-inflammatory activity.

**Table 1.** Summary of findings regarding circadian clock control of the NLRP3 inflammasome from murine models.

| Mice | Effect on Inflammation/NLRP3 Inflammasome | Day/Night Association | Possible Time-of-Day Effect on NLRP3 Inflammasome Activity | |
| --- | --- | --- | --- | --- |
| | | | Day | Night |
| **Circadian rhythms in hormonal regulators of inflammation** | At pharmacological concentrations, all are anti-inflammatory. Glucocorticoids can promote inflammasome priming | Higher circulating levels of glucocorticoids, melatonin and melanocortins at night | *Higher inflammatory cytokine production from active inflammasome* * | *Reduced inflammatory cytokine production from active inflammasome* * Increased susceptibility to inflammasome activation |
| **Time-of-day differences in expression of inflammasome components and responsiveness to TLR2 and TLR4 agonists** | Higher basal levels of inflammasome components may reduce threshold for inflammasome priming | High expression of *TLRs*, *NLRP3* and *RelA* and higher NF-κB activity at night | | Increased susceptibility to inflammasome activation |

* Effects that appear to be misaligned with a nighttime risk of gout flares are highlighted in italics.

**Table 2.** Summary of findings regarding circadian clock control of the NLRP3 inflammasome in humans and human macrophages.

| Humans | Effect on Inflammation/NLRP3 Inflammasome | Day/Night Association | Possible Time-of-Day Effect on NLRP3 Inflammasome Activity | |
| --- | --- | --- | --- | --- |
| | | | Day | Night |
| **Circadian rhythms in hormonal regulators of inflammation** | At pharmacological concentrations, all are anti-inflammatory. Glucocorticoids can promote inflammasome priming | Glucocorticoids and melanocortins high in morning; melatonin high at night | *Reduced inflammatory cytokine production from active inflammasomes throughout the day and night* * *Increased susceptibility to inflammasome-activating stimuli* * | |
| **Time-of-day differences in responsiveness of macrophages to MSU crystals** | Levels of caspase 1 activity and IL-1β production depend on time-of-day of MSU crystal exposure | Higher inflammatory response to MSU crystals at night? | | Greater extent of inflammation generated by MSU crystal-induced NLRP3 inflammasome activation |
| **Disruption of the circadian clock in macrophages by MSU crystal exposure** | Loss of circadian repression of *NLRP3* expression and NLRP3 inflammasome activation due to reduced levels of REV-ERBα and BMAL1 | Greater repression of REV-ERBα at night? BMAL1 naturally more abundant during the day? | | Increased susceptibility to inflammasome-activating stimuli Greater extent of inflammation generated by NLRP3 inflammasome |

* Effects that appear to be misaligned with a nighttime risk of gout flares are highlighted in italics. NOTE: Existing data on the day/night association of effects of MSU crystals on NLRP3 inflammasome activity in human macrophages a based on extrapolation from the timing of clock component cycling in an in vitro model.

Currently, data from both murine and human macrophages suggest that circadian control results in increased nighttime susceptibility to inflammatory stimuli and/or an

increased magnitude of inflammatory response generated by exposure to a stimulus at night.

*Limitations of Existing Studies*

Many of the existing data indicating a role for the circadian clock in the increased nighttime susceptibility to NLRP3 inflammasome activity in gout come from studies on nocturnal rodent models. Because of the inherent issue with potential circadian differences between nocturnal and diurnal animals, studies on human tissue are essential. To date, the only evidence of a time-of-day difference in NLRP3 inflammasome activity in response to MSU crystal exposure in human cells comes from studies on the THP-1 macrophage cell line. Determining whether the extent of NLRP3 inflammasome activation following MSU crystal exposure was greatest at night or during the day in these cells could only be based on extrapolation from the timing of clock protein abundance since the day/night context is lost in in vitro culture. There is a clear need to validate these findings in primary human cells and, importantly, to confirm whether there is an increased propensity for NLRP3 inflammasome activation in human macrophages at night. However, there are challenges associated with such studies.

Peripheral blood mononuclear cells (PBMCs) can easily be obtained from humans at different times of day and their responses to MSU crystal exposure can be compared over time. This would require minimal in vitro cell maintenance and provide the most direct assessment of in vivo responses possible. However, the factors which are under circadian clock control and the response of clocks to stimuli differ between different cell types, and therefore how PBMCs respond to MSU crystals may not necessarily mirror the response of other immune cells, such as differentiated macrophages. This is an important consideration, as while MSU crystal exposure was found to cause widespread changes in the expression of circadian clock components in PMA-differentiated THP-1 macrophages, the same was not evident in non-differentiated THP-1 monocytic cells [41]. Isolation of tissue-resident macrophages from humans is difficult, and therefore studies with human macrophages typically rely on in vitro differentiation of primary human PBMCs, a process requiring several days of in vitro culture [130,131]. Clocks in peripheral tissue cells rapidly become desynchronized in vitro, and although the circadian clock within each cell will continue to oscillate, neighboring cells can be at very different points in their circadian cycles at a given time [132,133]. This means that within a cell population, there will be a heterogenous mix of cells at different points in their circadian cycles, which is problematic for determining whether the response of cells to a stimulus differs depending on the time during the circadian cycle at which it is applied. A common method to overcome this is to re-synchronize cell clocks. The circadian clock and cell cycle are inextricably linked, and standard approaches used to re-synchronize cell clocks in vitro (treatment with dexamethasone or forskolin or serum shock) mirror those used to synchronize the cell cycle [133–135]. While these approaches are highly effective and enable the effects of stimuli applied at different times during the circadian cycle to be easily compared, they all affect the activity of multiple pathways in a cell aside from the circadian clock. This means that, using this approach, both circadian and non-circadian mechanisms could contribute to time-dependent differences in the response of cells to stimuli. It is crucial, therefore, that validation experiments (for example, knockdown/overexpression of clock components) are performed to demonstrate that time-of-day differences in circadian clock component abundance or activity contribute to time-of-day differences in responses. Of relevance in the context of understanding time-of-day differences in NLRP3 inflammasome activity, both dexamethasone and forskolin directly alter inflammatory signaling [42,136], confounding their use as re-synchronizing stimuli for circadian studies on inflammatory pathways. For this reason, serum shock was utilized in the studies showing time-of-day differences in the response of THP-1 macrophages to MSU crystals. While this study utilized pharmacological and gene knockdown/overexpression approaches to demonstrate that time-of-day differences in circadian clock component expression contributed to the time-

of-day differences in the response of THP-1 macrophages to MSU crystals, non-circadian pathways were likely also active and may have dampened or enhanced the magnitude of the differences observed. It is important that this limitation for circadian studies in cultured cells is recognized as it will also apply for studies with cultured primary human cells, such as PBMC-derived macrophages.

Aside from the issue of translating circadian studies from nocturnal rodent models to diurnal humans, there is also the limitation that most animal studies have involved animals maintained in a 12 h:12 h light/dark cycle and therefore animals with a 12 h:12 h sleep/wake cycle. In humans, the sleep/wake cycle is normally governed more by chronological time than the light/dark cycle. This means that the alignment between peripheral clock cycling (which can be dictated by rhythms in daily activity/behavior, such as mealtimes [23,24]) and central clock cycling might be quite different in a human compared to a laboratory animal. This is potentially relevant, since both central and peripheral clocks contribute to immune system control. Interestingly, in ulcerative colitis, a disease also associated with NLRP3 inflammasome activity and acute inflammatory flares, shiftwork and sleep disruption (both of which result in circadian disturbance and misalignment of central and peripheral clock cycling), are linked with increased risk of disease flares [137,138]. It is possible that misalignment between central and peripheral clocks increases susceptibility to NLRP3 inflammasome activation, and this may be relevant for understanding the risk of gout flares.

## 9. Circadian Involvement in Gout Flares in the Broader Context of Circadian Involvement in Inflammatory Diseases

Time-of-day differences in the onset of acute episodes of inflammation are common in inflammatory diseases. In humans, a distinct day/night patterning in symptom severity has been observed in a range of both infectious as well as chronic immune diseases, and this is at least partly attributed to circadian effects. Symptoms of influenza have been found to be more pronounced in the morning (0800–1100), and early morning stiffness is characteristic of chronic immune-mediated inflammatory arthritis, such as rheumatoid arthritis, and both are attributed to circadian control of the immune system [139,140]. Circadian rhythms in immune function are therefore a common mechanism leading to time-of-day differences in symptom severity in inflammatory disease. Differences in the timing of when symptoms are experienced following inflammatory pathway induction between diseases are likely a consequence of differences in the specific inflammatory pathways involved as well as differences in the magnitude of pain involved and whether it causes wakening from sleep. While NLRP3 inflammasome activity is central to gout flares, other pathways are more involved in driving early morning stiffness in RA [127,141]. Circulating levels of IL-6 are correlated with RA joint stiffness, and IL-6 levels start to rise in the early hours of the morning, peaking around the time of wakening in individuals with RA [142]. In contrast, the symptoms of gout flare typically develop in the early hours of the morning, but it is unknown how long before symptom onset NLRP3 inflammasome activation occurs, and inflammatory cytokine levels may start to rise hours before overt symptoms are experienced. It is possible that some circadian mechanisms, e.g., the nighttime nadir in circulating glucocorticoid levels may result in an increased propensity for inflammatory cytokine levels to build in the early morning hours, and this may be a shared mechanism contributing to the time-of-day patterning of disease symptoms across inflammatory diseases. However, it is likely that there are also some features of circadian control of inflammation that differ between diseases due to differences in the specific inflammatory pathways and specific immune cells most involved, as well as differences in the disease environment. This is in keeping with the current understanding that effects of the circadian clock on cell activity are both cell-type- and context-dependent. For instance, in the context of gout, MSU crystal-induced changes in expression of circadian clock components in macrophages could result in changes in circadian control of inflammatory pathway activity that are specific to gout and not seen in other diseases. It is interesting to note that NLRP3 inflammasome activation

is a central feature of ulcerative colitis, a disease also associated with acute inflammatory flares [143]. However, although there is evidence that circadian clock disruption could contribute to risk of colitis flare [137,138], there is currently no evidence of an increased risk of colitis flare at night [143].

## 10. Current Gout Flare Treatments and Circadian Regulation of Immune Function

The discovery of circadian rhythms in immune cell function has led to speculation that the efficacy of anti-inflammatory drugs could be optimized by coordinating the timing of administration with an individual's circadian clock. Low-dose colchicine, NSAIDs and glucocorticoids are recommended treatments for gout flare prophylaxis, while initiating urate-lowering therapy and full doses of these medications are recommended as first-line therapies in patients experiencing gout flare [87]. The possibility that time of day of administration influences the efficacy of these treatments has not been investigated in gout; however, this has been studied in relation to glucocorticoid use for RA. Several studies have demonstrated that nighttime administration of prednisolone results in a greater improvement in symptoms of early morning stiffness in individuals with RA compared to when administered in the morning [144,145]. This has been attributed to greater inhibition of the late-night acute increase in inflammation which is associated with early morning stiffness. However, aside from time-of-day differences in drug efficacy, time-of-day differences in adverse effects are also possible. This is particularly the case for synthetic glucocorticoids, which exert negative feedback on the hypothalamic–pituitary–adrenal (HPA) axis controlling endogenous glucocorticoid production [146]. The current recommendation for RA is to take glucocorticoids in the morning, and this is based on findings that synthetic glucocorticoids administered at 8 a.m. or 4 p.m. resulted in only temporary repression of endogenous glucocorticoid production, whereas when they were taken at 12 p.m., there was near full repression for 24 h [146]. These same considerations regarding potential adverse effects of synthetic glucocorticoids on the HPA axis also apply to gout flare.

Logically, drugs that might be expected to show marked differences in efficacy depending on time of day of administration are likely to be those with a short half-life that target a specific pathway which is strongly upregulated at a specific time of day. None of the drugs currently used for gout flare management are specific inhibitors of the NLRP3 inflammasome. For instance, the clinical efficacy of colchicine in gout flare management is attributed to a plethora of effects occurring at different stages in the inflammatory cascade. One of the major mechanisms by which colchicine is thought to exert anti-inflammatory activity is by inhibition of microtubule formation. This contributes to its inhibitory effect on the NLRP3 inflammasome [147]. The dissociation rate of colchicine from tubulin is very slow, with an estimated half-life of 20–30 h [148]. Therefore, even if there is a specific time of day at which inhibition of microtubule formation would be most effective in inhibiting inflammatory responses, the pharmacokinetics of colchicine may mean that it is more important that colchicine is taken at around the same time each day than the time of day at which it is taken. Inhibition of microtubule formation is not the only mechanism by which colchicine exerts anti-inflammatory effects, however, and it is possible that other aspects of colchicine activity may be influenced by the time of day of administration. This may also apply to the risk of colchicine toxicity, as colchicine is metabolized by cytochrome P450, which is also under circadian control [149].

There is some evidence that cyclooxygenase 2 (COX2) expression is regulated by the circadian clock [150], and one study found higher prostaglandin E2 (PGE2) levels in rodents during the period of light exposure compared to dark exposure [151]. Currently, however, there is limited evidence that time of day of NSAID treatment affects efficacy. While one study in mice showed that carprofen had greater efficacy with respect to post-operative healing and pain when administered during the normal hours of wakening than when administered during the rest period [152], a trial in humans found no effect of time of day of administration of ibuprofen for alleviation of pain following molar extraction [153].

Whether the time of day of NSAID administration influences the efficacy in gout flare treatment or prophylaxis is unknown.

Although the circadian clock regulates inflammation, inflammation also regulates the circadian clock. Exposure to inflammatory cytokines leads to altered expression of circadian clock components in immune as well as non-immune cells, and this is implicated as one of the mechanisms contributing to changes in the circadian clock within cells in a disease environment [154–156]. Exposure to anti-inflammatory drugs can also influence cell clocks. Of relevance, glucocorticoids and non-steroidal anti-inflammatory drugs (NSAIDs) which are used to treat the gout flare have been shown to regulate the circadian clock [134,152]. Whether this contributes to their clinical efficacy in gout flare management remains to be determined.

## 11. Conclusions and Future Direction

Existing data support the notion that circadian rhythms in immune cell function, driven by both central and peripheral circadian clocks, could contribute to time-of-day differences in susceptibility to NLRP3 inflammasome activation. Emerging evidence from studies in human macrophages indicates that exposure of macrophages to MSU crystals leads to changes in the expression of circadian clock components, leading to loss of normal circadian repression of NLRP3 inflammasome activation. This raises the possibility that disruption of the macrophage circadian clock contributes to the nighttime susceptibility to gout flares. However, there remain a number of unanswered questions that need to be addressed in order to understand the relative contribution of circadian clock control in the multi-factorial milieu of factors contributing to gout flares. These include:

- **determining whether circadian rhythms in circulating hormone levels contribute to time-of-day susceptibility to NLRP3 inflammasome-activating stimuli.** At present, there are no data demonstrating that physiologically relevant concentrations of melatonin or MSHs/ACTH can regulate the NLRP3 inflammasome, although data from pre-clinical models suggest that both may have efficacy in reducing inflammation when used pharmacologically. Similarly, knowledge regarding the effects of circadian rhythms in relation to endogenous glucocorticoid use and susceptibility to gout flares is also limited. In particular, the potential effect of glucocorticoids in promoting NLRP3 inflammasome priming has been largely unexplored in the context of gout. However, given that other nuclear receptors involved in regulating NLRP3 priming are also regulated by the circadian clock, this may be relevant for understanding the overall time-of-day susceptibility to NLRP3 inflammasome priming.

- **identifying whether there are time-of-day differences in the response of immune cells such as macrophages to stress-induced increases in endogenous levels of hormones, e.g., glucocorticoids.** Given that there are circadian rhythms in GR expression on target cells and that the profile of GR-regulated genes involved in energy metabolism has been shown to differ depending on the time of day of glucocorticoid exposure, it seems highly likely that the magnitude of the effect of stress-induced upregulation of glucocorticoid production on NLRP3 inflammasome activation and inflammation differs depending on the time of day. This may be relevant for understanding the link between stress and risk of gout flares.

- **confirming whether components of the NLRP3 inflammasome pathway are also expressed at higher levels in macrophages at night in humans, as has been observed in mice.** There is compelling evidence that the NLRP3 inflammasome is under circadian control in both murine and human macrophages. However, it is unclear whether circadian control of the inflammasome is the same in both species and, in particular, whether the day/night patterning is the same in diurnal humans as in nocturnal mice. Understanding this is critical for understanding the potential consequences of circadian control of the NLRP3 inflammasome for time-of-day susceptibility to activating stimuli.

- **establishing whether the changes in the expression of circadian clock components observed in MSU crystal-exposed THP-1 cells in vitro are also apparent in macrophages surrounding MSU crystal deposits within gout joints in vivo.** Although essential for understanding the translatability of in vitro findings, there are a number of barriers to performing this type of study in human tissue. The timing of tissue collection would need to be carefully considered, as would the type of tissue used as a comparator. In this regard, a combination of both human studies and studies in animal models may be necessary. The circadian clock is highly conserved across species, and therefore it is reasonable to use animal models to study factors that regulate the clock. While murine models have been extensively utilized for circadian clock research in other disease contexts, in relation to studies of gout flares, where there is a specific question about the day/night association of effects, diurnal models, for instance, zebrafish, may have greater utility.

- **determining how central and peripheral clock mechanisms interact to ultimately control NLRP3 inflammasome activity.** Circulating hormone levels controlled by the central clock as well as immune cell-specific peripheral clocks can regulate NLRP3 inflammasome activity. In addition, immune cell numbers oscillate over the course of a day, and circadian rhythms in phagocytosis have also been observed, both of which may also alter the overall extent of NLRP3 inflammasome activity in a joint. Understanding the interaction between the different levels of circadian control of immune cell function and inflammation as well as the impact of diurnal rhythms of exposure to inflammasome-activating stimuli (e.g., alcohol) is critical for ultimately understanding nighttime susceptibility to gout flares.

**Author Contributions:** Conceptualization, R.C.P. and N.D.; writing—original draft preparation, R.C.P.; writing—review and editing, R.C.P. and N.D. All authors have read and agreed to the published version of the manuscript.

**Funding:** This research received no external funding.

**Institutional Review Board Statement:** Not applicable.

**Informed Consent Statement:** Not applicable.

**Data Availability Statement:** No new data were created or analyzed in this study. Data sharing is not applicable to this article.

**Conflicts of Interest:** N.D. has received consulting fees, speaker fees or grants from AstraZeneca, Novartis, Dyve Biosciences, Horizon, Selecta, Arthrosi, JW Pharmaceutical Corporation, PK Med, LG Chem, JPI, PTC Therapeutics, Protalix, Unlocked Labs and Hikma outside of the submitted work. R.C.P. declares no conflict of interest.

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
