# Peer review of "Circadian Rhythms in NLRP3 Inflammasome Regulation: Possible Implications for the Nighttime Risk of Gout Flares"

_2813-4583, doi:10.3390/gucdd2020011_

Round 1

Reviewer 1 Report

Comments and Suggestions for Authors

The review article in question represents an amazing review in unravelling the phenomenon of the predilection for gout flares occurring predominantly late at night and in the early hours of the morning. The author's review of relevant studies not only illuminates this intriguing feature of gout but also sets the stage for further research in the field. This review serves as a foundational text for academics and clinicians alike, interested in the intersection of circadian biology and inflammatory pathology.

There are many interesting things to discuss, but I will focus my comments into the more mechanistic exploration of the circadian clock's interplay with myeloid cells, highlighting areas for enhancement and further exploration in the review.

Firstly, it's crucial to contextualize the circadian rhythm's involvement in gout within the broader spectrum of inflammatory diseases. Many such conditions exhibit acute inflammatory episodes at night, suggesting a common mechanism rather than a unique phenomenon to gout. The reviewer correctly points out that while gout may harbor some unique circadian signatures, the convergence on shared signaling pathways with other diseases is likely. This perspective could encourage a more holistic understanding of circadian biology in inflammatory responses, fostering cross-disease insights and therapeutic strategies.

The dissection of transcriptional and epigenetic landscapes, particularly in the context of monosodium urate crystals (MSUc)-stimulated inflammatory cells, is another highlight of the review. The citation of Cobo et al., 2022, shows the significant imprint of circadian clock proteins on the inflammatory cascade in the epigenetic changes induced by MSUc in non-primed macrophages. However, the review could further illuminate the necessity of these proteins to engage in a collaborative network with other transcription factors, forming a complex network that dictates a particular, cell-specific and context-dependent transcriptional output. This transcriptional regulation extends beyond the circadian components, acknowledging the intricacies of cellular machinery during the response of macrophages to MSUc. For instance, the DNA motifs recognized by circadian clock proteins exhibit notable similarities to those bound by basic helix-loop-helix (bHLH) proteins and certain nuclear receptors, providing some molecular insights into the transcriptional regulation of circadian activity. This resemblance hints at a sophisticated interplay between circadian regulators and a broader array of transcription factors, underscoring the complexity and integration of circadian control within cellular regulatory networks.

The critique of in vitro models, especially the prevalent use of THP1 cells for studying circadian influence on inflammation, introduces critical considerations. The deviation of these models from primary human macrophage biology, particularly under conditions of PMA-induced differentiation, poses significant limitations to our understanding of circadian dynamics in gout and related inflammatory diseases. The review would benefit from a discussion on the imperative of corroborating findings in THP1 cells with human primary cells and in vivo studies, to transcend the limitations posed by these models. The review would also benefit from a discussion of how the serum-shock model of synchronization could affect the interpretation of the obtained data.

Furthermore, the review's focus on inflammasome activation in the context of circadian regulation could be expanded to encompass the upstream regulatory mechanisms of phagocytosis. The involvement of circadian clock proteins in modulating phagocytic activity, as suggested by several studies (PMID 17409491, PMID: 31900362, PMID: 28087238, among others), calls for a discussion of the topic. This not only enriches the narrative on inflammasome activation but also aligns with the pathogenesis narrative of gout flares, where the uptake of crystal particles by phagocytic cells is central.

Lastly, an exploration of how conventional gout flare treatments intersect with circadian regulation could unveil novel therapeutic avenues. Given the influence of the circadian clock on inflammatory diseases, there exists a new avenue to amplify the efficacy of existing drugs through circadian modulation. This foresight could inspire translational research endeavors aimed at optimizing treatment regimens for gout and potentially other circadian-influenced inflammatory conditions.

In conclusion, while the review provides an amazing exploration of the circadian clock's role in gout flare timing, it will benefit, in my opinion, from a deeper mechanistic insight and broader contextualization within the frame of inflammatory diseases. This reviewer believes that addressing the highlighted considerations could significantly enrich the discourse, paving the way for groundbreaking discoveries in circadian biology and its therapeutic exploitation in gout and beyond, perhaps even looking at the contribution of gout and other comorbidities such as cardiovascular disease.

Author Response

We thank the Reviewers for their time taken in reviewing our manuscript and their thoughtful comments on our Review. We have addressed each comment raised by Reviewers, incorporating three new sections into our manuscript. Please find below a point-by-point explanation of the changes made to our manuscript to address each Reviewer comment.

Reviewer 1 

The review article in question represents an amazing review in unravelling the phenomenon of the predilection for gout flares occurring predominantly late at night and in the early hours of the morning. The author's review of relevant studies not only illuminates this intriguing feature of gout but also sets the stage for further research in the field. This review serves as a foundational text for academics and clinicians alike, interested in the intersection of circadian biology and inflammatory pathology.

There are many interesting things to discuss, but I will focus my comments into the more mechanistic exploration of the circadian clock's interplay with myeloid cells, highlighting areas for enhancement and further exploration in the review.

  1. Firstly, it's crucial to contextualize the circadian rhythm's involvement in gout within the broader spectrum of inflammatory diseases. Many such conditions exhibit acute inflammatory episodes at night, suggesting a common mechanism rather than a unique phenomenon to gout. The reviewer correctly points out that while gout may harbor some unique circadian signatures, the convergence on shared signaling pathways with other diseases is likely. This perspective could encourage a more holistic understanding of circadian biology in inflammatory responses, fostering cross-disease insights and therapeutic strategies.

We have added an additional section to the Review to address this topic (“Circadian involvement in the gout flare in the broader context of circadian involvement in inflammatory diseases” (section 8, page 17, lines 677-714). While we agree that it is likely there are some common circadian mechanisms underlying time of day differences in acute inflammatory episodes across diseases (e.g. the nighttime nadir in circulating glucocorticoid levels likely creates a nighttime vulnerability to inflammatory cytokine accumulation), we also think there are some aspects of circadian control which may be unique to specific diseases. This is in keeping with our understanding that the activity of circadian clocks is both cell type and context specific (as highlighted by the Reviewer in the point below.). In particular, findings that MSU crystals cause changes in the circadian clock in macrophages and that this influences NLRP3 inflammasome activity suggests that there may also be some aspects of circadian control of NLRP3 inflammasome activation that are unique to gout.   

  1. The dissection of transcriptional and epigenetic landscapes, particularly in the context of monosodium urate crystals (MSUc)-stimulated inflammatory cells, is another highlight of the review. The citation of Cobo et al., 2022, shows the significant imprint of circadian clock proteins on the inflammatory cascade in the epigenetic changes induced by MSUc in non-primed macrophages. However, the review could further illuminate the necessity of these proteins to engage in a collaborative network with other transcription factors, forming a complex network that dictates a particular, cell-specific and context-dependent transcriptional output. This transcriptional regulation extends beyond the circadian components, acknowledging the intricacies of cellular machinery during the response of macrophages to MSUc. For instance, the DNA motifs recognized by circadian clock proteins exhibit notable similarities to those bound by basic helix-loop-helix (bHLH) proteins and certain nuclear receptors, providing some molecular insights into the transcriptional regulation of circadian activity. This resemblance hints at a sophisticated interplay between circadian regulators and a broader array of transcription factors, underscoring the complexity and integration of circadian control within cellular regulatory networks.

We have expanded our discussion on the mechanisms by which cell clocks affect cell activity and the interplay between circadian clocks and other regulatory mechanisms in a cell. We have broadened this discussion to highlight that the circadian clock contributes to the transcriptional and epigenetic landscape within a cell as well as contributing to the control of protein translation. The latter has received much less research attention than transcriptional control mechanisms but recent findings in macrophages that only 15% of the circadian proteome show circadian oscillations at the RNA level indicates that translational control may be particularly relevant for circadian control of immune function.  Discussion added to Page 4, lines 133-153. 

  1. The critique of in vitro models, especially the prevalent use of THP1 cells for studying circadian influence on inflammation, introduces critical considerations. The deviation of these models from primary human macrophage biology, particularly under conditions of PMA-induced differentiation, poses significant limitations to our understanding of circadian dynamics in gout and related inflammatory diseases. The review would benefit from a discussion on the imperative of corroborating findings in THP1 cells with human primary cells and in vivo studies, to transcend the limitations posed by these models. The review would also benefit from a discussion of how the serum-shock model of synchronization could affect the interpretation of the obtained data.

We have added a new section to the Review titled “Limitations of existing studies” (section 7.1; page 15, line 606 - page 17, line 675). Here we discuss limitations of in vitro approaches including why the serum shock model (as well as other models for synchronising cell clocks) are used and the limitations associated with those approaches. We also highlight the need for studies in primary human cells and highlight some considerations and limitations of those studies and also discuss limitations of current in vivo animal model studies.

  1. Furthermore, the review's focus on inflammasome activation in the context of circadian regulation could be expanded to encompass the upstream regulatory mechanisms of phagocytosis. The involvement of circadian clock proteins in modulating phagocytic activity, as suggested by several studies (PMID 17409491, PMID: 31900362, PMID: 28087238, among others), calls for a discussion of the topic. This not only enriches the narrative on inflammasome activation but also aligns with the pathogenesis narrative of gout flares, where the uptake of crystal particles by phagocytic cells is central.

There are many other aspects of circadian control of the immune system upstream to NLRP3 inflammasome activation which can influence the overall sensitivity of the inflammasome to stimuli and magnitude of response generated. These include phagocytosis but also circadian rhythms in immune cell numbers. We highlighted thatcircadian rhythms in immune cell numbers likely also influenced inflammasome activity in our original manuscript (first mentioned Page 4, line 161 and further discussed in Conclusions, page 21, lines 832-833). We also mentioned that phagocytosis is under circadian control in our original manuscript (page 4, line 160 but have now expanded our discussion on this (Page 11, lines 439-445) and also highlighted its relevance in the Conclusions (Page 21, line 833). There is currently very little data on circadian rhythms in phagocytosis in human cells and this is something we now highlight as a gap in the literature given the relevance of phagocytosis for gout flare. 

  1. Lastly, an exploration of how conventional gout flare treatments intersect with circadian regulation could unveil novel therapeutic avenues. Given the influence of the circadian clock on inflammatory diseases, there exists a new avenue to amplify the efficacy of existing drugs through circadian modulation. This foresight could inspire translational research endeavors aimed at optimizing treatment regimens for gout and potentially other circadian-influenced inflammatory conditions.

In keeping with comments from both Reviewers we have included a new section in our Review titled “Current gout flare treatments and circadian regulation of immune function (Section 9, Page 17 line 716 – page 19, line 774).

In conclusion, while the review provides an amazing exploration of the circadian clock's role in gout flare timing, it will benefit, in my opinion, from a deeper mechanistic insight and broader contextualization within the frame of inflammatory diseases. This reviewer believes that addressing the highlighted considerations could significantly enrich the discourse, paving the way for groundbreaking discoveries in circadian biology and its therapeutic exploitation in gout and beyond, perhaps even looking at the contribution of gout and other comorbidities such as cardiovascular disease.

Reviewer 2 Report

Comments and Suggestions for Authors

Authors review in this work the control of the inflammasome NLRP3 by the circadian clock and the impact of such regulation in the context of gout flare. Overall, the manuscript is very-well written and organized using an almost appropriate literature. I particularly appreciated the historical introduction, which often lacks in many current reviews. This work is timely and should be of interest to a specialized audience such as the reader ship of GUCDD. We believe that the following remarks should be addressed to improve comprehensiveness, relevance and accuracy of the manuscript.

1. A schema integrating the NLRP3 activation processes (priming and activation) with an emphasis on MSU would help non-NLRP3 readers to understand these pathways.

2. Regarding the direct regulation of NLRP3 expression, neither Wang et al Nat Com (late dec 2018) nor Popov D et al (2023) demonstrate that NR1D1/Rev-erb-alpha control NLRP3 in the liver. Instead, the first paper describing the circadian expression of NLRP3 and reporting the direct molecular mechanisms by which Rev-erb-alpha regultes the circadian expression of NLRP3 is Pourcet et al, Gastroenterology 2018 ePub dec 2017. Strikingly, this article demonstrates for the first time that Rev-erb-alpha controls NLRP3 circadian expression in vivo in peritoneal macrophage, in liver, in vitro in mouse BMDM and in human primary macrophages, which is a much more relevant model than THP1 cell line. In this article, two in vivo models were used: peritonitis and LPS/GalN-induced fulminant hepatitis. Citation of this pioneer article is then missing.

3. More than glucocorticoids and GR, many other transcription factors, including nuclear receptors, have been involved in the control of the priming and the activation step of Nlrp3. This should be developed as well.

4. Clinical relevant studies relative to night gout flare and/or NLRP3 activity should be mentioned/discussed. In addition, it may be interested to discuss current treatment used to treat gout including colchicine, thought to inhibit, Nlrp3. Although colchicine also controls microtubule stability, it may be interested to place such treatment in the perspective of the review.

Author Response

We thank the Reviewers for their time taken in reviewing our manuscript and their thoughtful comments on our Review. We have addressed each comment raised by Reviewers, incorporating three new sections into our manuscript. Please find below a point-by-point explanation of the changes made to our manuscript to address each Reviewer comment.

Reviewer 2

Authors review in this work the control of the inflammasome NLRP3 by the circadian clock and the impact of such regulation in the context of gout flare. Overall, the manuscript is very-well written and organized using an almost appropriate literature. I particularly appreciated the historical introduction, which often lacks in many current reviews. This work is timely and should be of interest to a specialized audience such as the reader ship of GUCDD. We believe that the following remarks should be addressed to improve comprehensiveness, relevance and accuracy of the manuscript.

  1. A schema integrating the NLRP3 activation processes (priming and activation) with an emphasis on MSU would help non-NLRP3 readers to understand these pathways.

We have added a schema to the manuscript showing the NLRP3 inflammasome activation process and highlighting the role of MSU crystals. (Figure 2, Page 6).

  1. Regarding the direct regulation of NLRP3 expression, neither Wang et al Nat Com (late dec 2018) nor Popov D et al (2023) demonstrate that NR1D1/Rev-erb-alpha control NLRP3 in the liver. Instead, the first paper describing the circadian expression of NLRP3 and reporting the direct molecular mechanisms by which Rev-erb-alpha regultes the circadian expression of NLRP3 is Pourcet et al, Gastroenterology 2018 ePub dec 2017. Strikingly, this article demonstrates for the first time that Rev-erb-alpha controls NLRP3 circadian expression in vivo in peritoneal macrophage, in liver, in vitro in mouse BMDM and in human primary macrophages, which is a much more relevant model than THP1 cell line. In this article, two in vivo models were used: peritonitis and LPS/GalN-induced fulminant hepatitis. Citation of this pioneer article is then missing.

We have now included the citation to the Pourcet study (reference #110; Page 11, line 477 and Page 11 line 482 – page 12 line 485). This was one of the papers we reviewed as it is a key paper in the field, and we apologize that we inadvertently missed adding the reference to this paper in our citation list. We thank the Reviewer for noticing this error.

  1. More than glucocorticoids and GR, many other transcription factors, including nuclear receptors, have been involved in the control of the priming and the activation step of Nlrp3. This should be developed as well.

We have expanded discussion on this. To maintain the focus of the review on circadian control of the NLRP3 inflammasome, we have limited our discussion to other regulators of NLRP3 priming that are also believed to be under circadian control. This discussion has been added on Page 9, lines 349-356. 

  1. Clinical relevant studies relative to night gout flare and/or NLRP3 activity should be mentioned/discussed. In addition, it may be interested to discuss current treatment used to treat gout including colchicine, thought to inhibit, Nlrp3. Although colchicine also controls microtubule stability, it may be interested to place such treatment in the perspective of the review.

In keeping with comments from both Reviewers we have included a new section in our Review titled “Current gout flare treatments and circadian regulation of immune function (Section 9, Page 17 line 716 – page 19, line 774). This includes discussion of colchicine along with glucocorticoids and NSAIDs.

Round 2

Reviewer 2 Report

Comments and Suggestions for Authors

I would like to thanks authors to have address my main concerns.

I would suggest, in the new figure 2, to add in the priming step, an alternative priming pathway apart from NF-kB pathway, as mentioned in my previous comments. Indeed, many other transcription factors, including Bmal1, ROR and Rev-erb, which belong to the clock, directly control Nlrp3 component pathways. Mechanism involving Bmal1 and ROR were not mentioned either while, based on authors comments, they are directly linked to the clock. This should be addressed.

Finally, As the ref 110 (Pourcet et al, is the first work demonstrating the direct circadian regulation of Nlrp3 by REV-ERBa using ChIP experiment in macrophages, it should be cited altogether with ref 109 in p11, lane 475 and 477. The term darkness should be used in the this context. Instead, authors should use rest period or activity period, as it might be confusing for a readership uninitiated to the circadian field.

As mentioned for CLOCK, Rev-erb-a has also been shown to indirectly regulate NLRP3 via NF-kB (Wang et al ref 109). This should be mentioned in page 12 as well.

Author Response

I would suggest, in the new figure 2, to add in the priming step, an alternative priming pathway apart from NF-kB pathway, as mentioned in my previous comments. Indeed, many other transcription factors, including Bmal1, ROR and Rev-erb, which belong to the clock, directly control Nlrp3 component pathways. Mechanism involving Bmal1 and ROR were not mentioned either while, based on authors comments, they are directly linked to the clock. This should be addressed.

We have added to the figure to acknowledge that there are alternative priming pathways independent of TLR and/or NF-κB. This comment has also been added to the manuscript text (page 5, lines 182-185.)  We have not included specific details of these pathways and regulators in the figure as that would make the figure too complex for easy interpretation. We added discussion regarding AhR specifically in our previous revision (page 9, lines 356 -358) because expression of this particular regulator has been shown to vary over the course of the day therefore discussion of AHR is directly relevant to the topic of this review. To our knowledge, the mechanism by which BMAL1 regulates NLRP3 expression is unknown. We have added a statement regarding this on page 11, lines 487-489. ROR-γ has been shown to directly bind to the promoter regions of both NLRP3 and IL-1β to promote their expression. We have also added a sentence to describe this (page 11, lines 489-491).  

Finally, As the ref 110 (Pourcet et al, is the first work demonstrating the direct circadian regulation of Nlrp3 by REV-ERBa using ChIP experiment in macrophages, it should be cited altogether with ref 109 in p11, lane 48a and 477. The term darkness should be used in the this context. Instead, authors should use rest period or activity period, as it might be confusing for a readership uninitiated to the circadian field.

The Pourcet reference has been added (page 11, lines 482 and 484 in revised manuscript). We have added a comment to state that the hours of darkness relate to the active period for a nocturnal mouse (Page 11, lines 481-482) but believe it is important to keep the reference to darkness/light since the light/dark cycle directly regulates the clock whereas rest/activity rhythms are an output of the clock. In the studies we discuss, the light/dark cycle was directly controlled not rest/activity rhythms and while the two would be expected to be linked in laboratory animals, it does depend how the animals were maintained (e.g. if cage washing/feed replenishment etc is being undertaken during the hours of light , those animals are very unlikely to be fully entrained to the light/dark cycle). No measure of rest/activity rhythms in most of these studies,  therefore for accuracy we have maintained reference to light/dark. At present it is unknown whether rhythms in TLR, NLRP3, NF-kB are linked to the light/dark cycle or are driven by peripheral clocks (which could be influenced by rest/activity rhythms) or a combination of both. We highlight this specific issue with extrapolating studies from nocturnal animals to diurnal humans in our manuscript (Page 12, lines 504-517 and Page 16, line 666 – Page 17, line 680).    

As mentioned for CLOCK, Rev-erb-a has also been shown to indirectly regulate NLRP3 via NF-kB (Wang et al ref 109). This should be mentioned in page 12 as well.

In our manuscript we stated REV-ERBα represses transcription of RelA. We have now further modified this sentence to state that the effect is through direct control and to make it explicit that this also contributes to the mechanism by which regulates NLRP3 expression.  (Page 11 (lines 492-493).